# On Sea-Level Change in Coastal Areas

**Vincent Courtillot [1], Jean-Louis Le Mouël [1], Fernando Lopes [1,\*] and Dominique Gibert [2]**

1   Institut de Physique du globe de Paris, Université de Paris, 75005 Paris, France
2   LGL-TPE, Université de Lyon 1, ENSL, CNRS, UMR 5276, 69622 Villeurbanne, France
\*   Correspondence: lopesf@ipgp.fr

**Abstract:** Variations in sea-level, based on tide gauge data (**GSLTG**) and on combining tide gauges and satellite data (**GSLl**), are subjected to singular spectrum analysis (**SSA**) to determine their trends and periodic or quasi-periodic components. **GLSTG** increases by 90 mm from 1860 to 2020, a contribution of 0.56 mm/yr to the mean rise rate. Annual to multi-decadal periods of ∼90/80, 60, 30, 20, 10/11, and 4/5 years are found in both **GSLTG** and **GSLl**. These periods are commensurable periods of the Jovian planets, combinations of the periods of Neptune (165 yr), Uranus (84 yr), Saturn (29 yr) and Jupiter (12 yr). These same periods are encountered in sea-level changes, the motion of the rotation pole **RP** and evolution of global pressure **GP**, suggesting physical links. The first SSA components comprise most of the signal variance: 95% for **GSLTG**, 89% for **GSLl**, 98% for **GP** and 75% for **RP**. Laplace derived the Liouville–Euler equations that govern the rotation and translation of the rotation axis of any celestial body. He emphasized that one must consider the orbital kinetic moments of all planets in addition to gravitational attractions and concluded that the Earth's rotation axis should undergo motions that carry the combinations of periods of the Sun, Moon and planets. Almost all the periods found in the **SSA** components of sea-level (**GSLl** and **GSLTG**), global pressure (**GP**) and polar motion (**RP**), of their modulations and their derivatives can be associated with the Jovian planets. The trends themselves could be segments of components with still longer periodicities (e.g., 175 yr Jose cycle).

**Keywords:** sea level; tide gauges; global sea pressure; mean pole path

## 1. Introduction

A global rise in sea-level has become a topic of major concern for the both inhabitants of coastal areas and regions of very low altitude and a huge amount of studies have been devoted to the description and understanding of evolutions in global sea-level. The problem is sometimes confusing because the definition of sea-level rise (and drop) is not always clear and unique and the observations on which it is based have been obtained by very different methods (tide gauges, GPS measurements, satellite observations). Referring to works by [1–6], we define absolute the sea-level **SL**(P,t) as the difference between (1) the distance from the center of mass of the Earth to the sea surface, at time t and location P, and (2), the distance from the center of mass of the Earth to the sea bottom (solid Earth), which is the water depth, at the same time t and location P:

$$\mathbf{SL}(P,t) = R_{ss}(P,t) - R_{se}(P,t) \tag{1}$$

The evolution of sea level between time t and a reference time $t_r$ is simply:

$$\begin{aligned}\mathbf{SLE}(P,t) &= \mathbf{SL}(P,t) - \mathbf{SL}(P,t_r) \\ &= [R_{ss}(P,t) - R_{se}(P,t)] - [R_{ss}(P,t_r) - R_{se}(P,t_r)]\end{aligned} \tag{2}$$

which can be rearranged as:

$$\mathbf{SLE}(P,t) = [R_{ss}(P,t) - R_{ss}(P,t_r)] - [R_{se}(P,t) - R_{se}(P,t_r)] \tag{3}$$

In that form, changes in sea level are the difference between the changes in altitude of the sea surface (i.e., the geoid), which are measured by tide gauges in coastal areas and changes in the altitude of the solid Earth that can be measured at GPS stations (e.g., [7,8]). The geoid term $[R_{ss}(P,t) - R_{ss}(P,t_r)]$ is modeled as: $\frac{G}{\gamma}(P,t) + c(t)$, where G is the full gravitational potential G gravity at the Earth surface and $c(t)$ stands for changes in masses at the surface, including ice melting [9].

Despite the actual complexity of the consequences of the (apparently simple) law of attraction of masses, Laplace [10] was able to build a full theory of celestial mechanics. He wrote a whole volume (the book IV of their Treatise) on the oscillations of the sea and of the atmosphere. Laplace showed that, in order for sea-level to be in equilibrium, the sum of forces had to be zero, eventually resulting in the famous Liouville–Euler system of partial differential equations. Tides were understood to be second-order oscillations, whose phases and amplitudes could be computed. The mathematical stage was set for the study of changes in sea-level.

The early 1930s saw a period of active interest in the determination of sea level and its changes, and the physical causes of these changes. Nomitsu and Okamoto [11] attributed **SL** variations in the Sea of Japan to variations in density and atmospheric pressure. Marmer [12] noted an annual oscillation at the tide gauges of Seattle and San Francisco that he also linked to meteorological phenomena. After a decade of additional observations, Jabobs [13] and LaFond [14] concluded that there was indeed a general rise in sea-level. Several explanations were considered: insulation from the Sun, meteorological tides, water density, changes in geography and geology of ocean basins. None was found to be completely satisfactory. McEwen [15] envisioned a complex mechanism with evaporation in the summer and precipitation in winter affecting the global sea-level. Further study showed that these annual variations were in fact in the phase in all observation stations, and the idea had to be dropped. Only the pressure patterns and hence the winds remained as a potential cause of periodic **SL** variations.

By the end of the 1970s and early 1980s, the principal cause of sea-level rise came to be attributed to a major phase of warming having resulted in the melting of the northern hemisphere ice caps. Refs. [16,17] elaborated on the theory of post-glacial isostatic rebound. The global warming of the atmosphere that took place over the past 150 years or so was attributed to anthropic release of greenhouse gases, contributing to the melting of glaciers and a rise in **SL** [18–25].

Our knowledge of sea-level and its variations in the 19th century and up to the present is primarily based on observations made by tide gauges. These provide the first half of Equation (3), that is $[R_{ss}(P,t) - R_{ss}(P,t_r)]$. One of the longest series comes from the Brest (France) tide gauge, established in 1807 and still in function. Le Mouël et al. [26] recently analyzed the Brest data and in parallel Earth's pole positions (from the IERS, 1845–2019), submitting both of them to singular spectral analysis (**SSA**). The first **SSA** components of both series, i.e., the trends, are very similar, with a major acceleration event near 1900 and a sea-level lagging pole motion by 5–10 years. **SSA** components with periods 1 yr, ∼11 yr and 5.4 yr are common to the two series. An important feature is a 0.5 yr component that is present in sea-level but absent from pole motion. The remarkable similarity of the two trends and their phase lag suggests a causal relationship opposite to what is generally accepted.

The primary aim of the present paper is to apply the same analysis to the global database of tide gauges. The tide gauge data are maintained by the Permanent Service of Mean Sea Level (**PSMSL**, https://www.psmsl.org/data/obtaining/complete.php (accessed on 5 July, 2020). Measurements of coastal sea level are available at 1548 sites; the raw data at all sites are shown in Figure 1.

The second half of Equation (3), that is, vertical land motion **VLM** = $[R_{se}(P,t) - R_{se}(P,t_r)]$, can be accessed through GPS measurements. These allow far better coverage of Earth surface but cover a much shorter span of time, generally 30 yr at most. For that reason, only "recent" trends covering that period can be accessed. A very thorough analysis of

these data has recently been published by [8]. The database is maintained by the Nevada Geodetic Laboratory, with 19286 sites scattered throughout the globe, and is called Median Interannual Difference Adjusted for Skewness (**MIDAS**, http://geodesy.unr.edu/velociti es/ (accessed on 5 July 2020).

Since the beginning of the 1990s, the sea level has also been monitored by series of altimetric satellites that have allowed much denser global coverage but (as is the case for GPS measurements) over a short time range of 30 years. A number of recent papers have attempted to combine these very different data sets, resulting in various global sea-level (**GSL**) curves (e.g., [8,27]). Interest in these **GSL** curves has focused on their trends, acceleration, annual and inter-annual variations, and on the mechanisms responsible for these variations.

We recently successfully applied the **SSA** method (see Golyandina and Zhigljavsky [28], Lemmerling and Van Huffel [29], Golub and Reinsch [30]) to a number of geophysical and heliophysical time series. **SSA** decomposes any time series into a sum of components, a trend (that may or may not be present) and stationary quasi-periodic components. We have explained the method in a number of papers [31,32]. For instance, oscillations (pseudo-cycles) of ~160, ~90, ~60, ~22 and ~11 yr are found in a series of sunspot numbers (e.g., Refs. [33–39]) as well as in a number of terrestrial phenomena [31,32,40–53], particularly sea level [26,54–57]. These particular periods (or periodicities) are of special interest, as they are members of the family of commensurable periods of the Jovian planets acting on the Earth and Sun [39,41,49,51,58]. These values are indeed close to the revolution periods of Neptune (165 yr), Uranus (84 yr), Saturn (29 yr) and Jupiter (12 yr) and several of their commensurable periods (see Table 1 in [51]).

In Section 2 of this paper, we discuss the tide gauge records and perform an **SSA** of these. In Section 3, we discuss and analyze the time series of vertical land motion (**VLM**) based on GPS measurements. In Section 4, we submit some global sea-level (**GSL**) curves that include satellite observations to **SSA**. We compare and discuss the results of these analyses in Section 5 and conclude in Section 6.

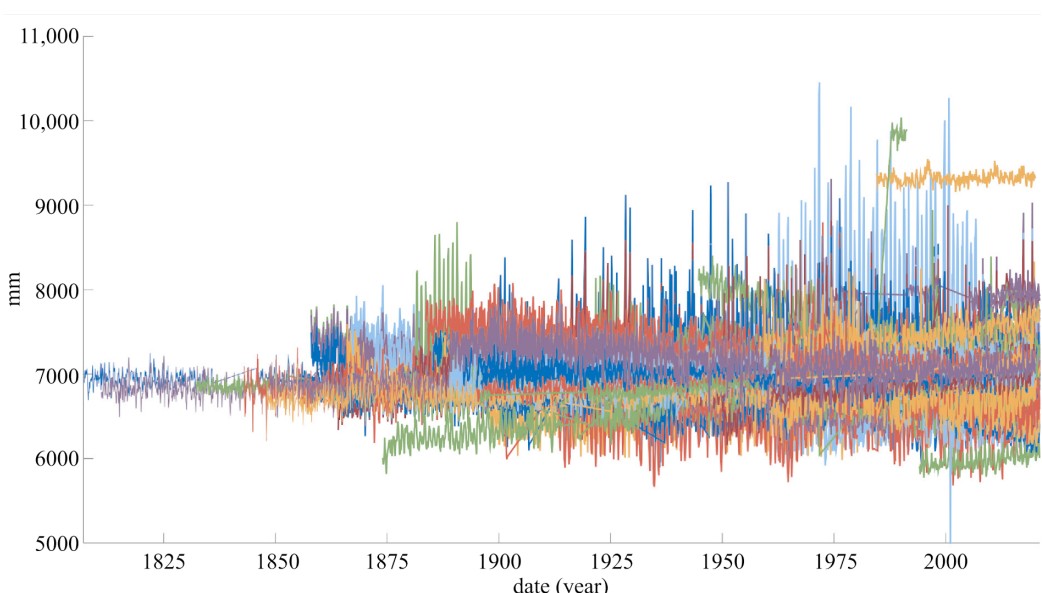

**Figure 1.** The full tide gauge database (1548 tide gauges of **PSMSL** series).

## 2. SSA of Tide Gauge Records Time Series (i.e., $[R_{ss}(P,t) - R_{ss}(P,t_r)]$)

Drawing on a recent paper [26], in which we analyzed the record from the Brest tide gauge since 1807, using SSA to extract its main components, we proposed to build a global sea-level curve covering the past 200 years using data from tide gauges only (**GSLTG**). We are aware of the uncertainties due to geography and tectonics but are interested in subjecting the original data to as little manipulation as possible.

We have selected the recordings from 31 stations (Table 1) on the basis of two criteria: the longest possible span of time and the best possible—if still too restrained—spatial coverage (Figure 2).

**Table 1.** List of tide gauges selected in this study to build the **GSLTG** series.

| Tide Gauge Site | Lat (°) | Lon (°) |
|---|---|---|
| Adak Sweeper Cove | 51.863 | −176.632 |
| Argentine Islands | −65.246 | −64.257 |
| Auckland | −36.843 | 174.769 |
| Brest | 48.382 | 4.494 |
| Churchill | 58.767 | −94.183 |
| Chennai | 13.100 | 80.300 |
| Cochin | 9.967 | 76.267 |
| East-London | −33.027 | 27.932 |
| Fernandina Beach | 30.672 | −81.465 |
| Honolulu | 21.307 | −157.867 |
| Ketchikan | 55.332 | −131.625 |
| Key West | 24.555 | −81.806 |
| Knysna | −34.049 | 23.046 |
| Lisbon | 38.700 | −9.133 |
| Marseille | 43.279 | 5.354 |
| Maasluis | 51.918 | 4.25 |
| Manila, S.harbor | 14.583 | 120.967 |
| Mera | 34.919 | 139.825 |
| Montevideo | −34.900 | −56.250 |
| Narvik | 68.428 | 17.426 |
| Oslo | 59.909 | 10.735 |
| Polyarny | 69.200 | 33.483 |
| Port Adelaide | −34.780 | 138.481 |
| Rio de Janeiro | −22.933 | −43.133 |
| Swinoujscie | 53.917 | 14.233 |
| Takoradi | 4.885 | −1.745 |
| Tofino | 49.150 | −125.917 |
| Tuapse | 44.100 | 39.067 |
| Valparaiso | −33.027 | −71.626 |
| Visby | 57.639 | 18.284 |
| Xiamen | 24.450 | 118.067 |

We have also performed the analyses that follow on the full database (1548 tide gauge stations): this results in increased dispersion without changing what can be drawn from the selection of 31 stations. The raw data from the 31 stations are all displayed without any filtering or processing, except for the fact that the first value has been subtracted, so that all series begin with a zero and only relative vertical variations of **SL** are displayed (Figure 3).

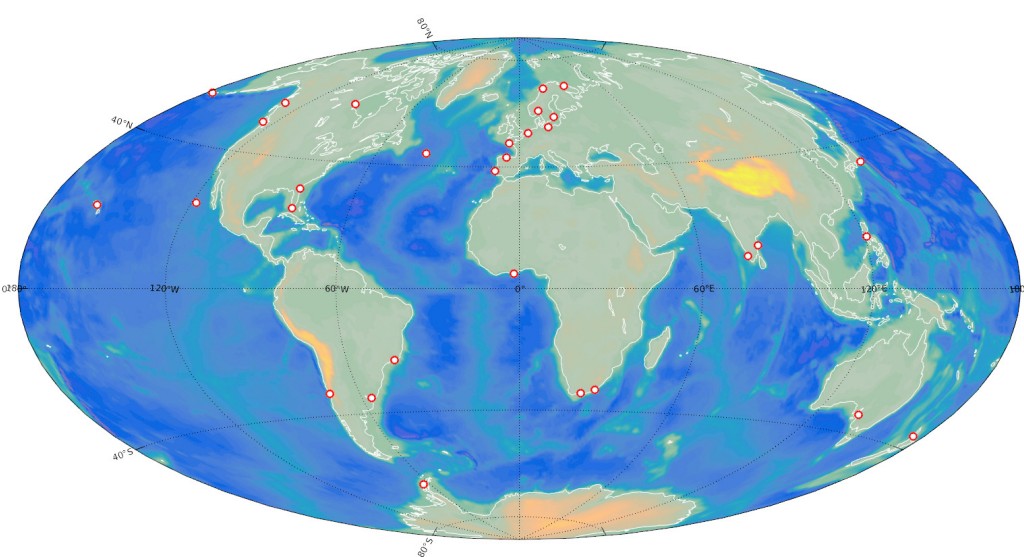

**Figure 2.** Map showing the locations of tides gauges used to build the **GSLTG** series.

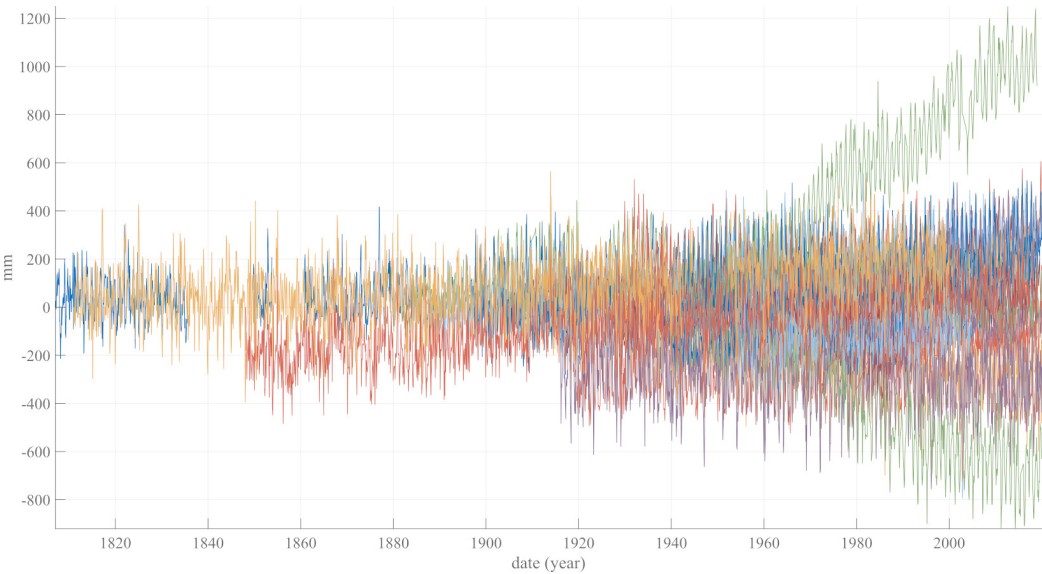

**Figure 3.** Superposition of data series from the 31 tide gauges listed in Table 1.

The two tide gauges that have large slopes (plotted in green in Figure 3), one positive (Manila, S. Harbor, Philippines) and one negative (Churchill, Canada), outline a funnel shape (see Table 1). The funnel undergoes rapid growth after 1950, with a total amplitude of 2 m, determined by the two extrema at Manila and Churchill. The annual oscillation, first identified by Marmer [12], has an amplitude in the order of 200 mm (peak to trough) for all tide gauges. The individual data points are shown in Figure 4a.

We then determine a smooth mathematical model of the mean sea-level variations as a function of time, in a least squares sense (e.g., [59]). For a mathematical representation of this smooth model, we can choose between polynomials, sine functions, exponentials, or splines. We select sine functions, based on our previous experience with the Brest data [26]. We add sine components progressively, until no more significant information is brought by the new component (in the sense of the Akaike criterion, see [60] used in [45]). In the present case, this leads to a model consisting of eight sine components. This model is shown with the data in Figure 4a and enlarged in Figure 4b.

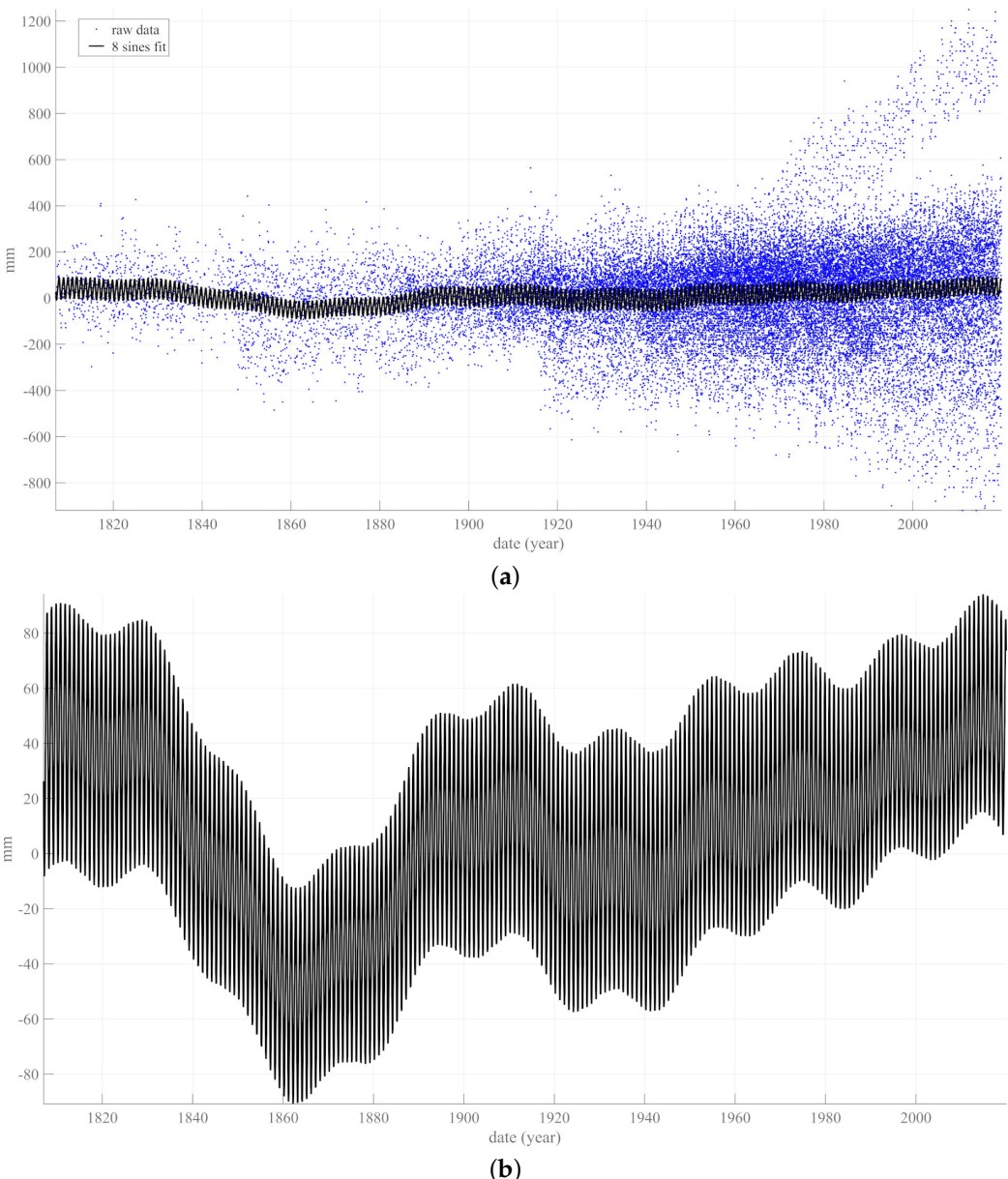

**Figure 4.** Fit of the 31 tide gauges. (**a**) Data points from the 31 tide gauges used in building **GSLTG** and their sinusoidal fit, with 8 sine waves. (**b**) The fit (black curve) of (**a**) with an enlarged ordinate scale.

The model of Figure 4a represents the mean variation common to all 31 tide gauges. At first glance, it consists of an annual oscillation, with a peak to trough amplitude of approximately 100 mm and a rather constant overall mean value. Figure 4b shows more precisely the modulations of this amplitude. The remaining scatter of data points can be assigned to local or regional variability. We call this model **GSLTG**. The **GSLTG** curve is then analyzed using **SSA**. The analysis yields a sum of components with (pseudo-) periods of 1 yr, 90 yr, 60 yr, 80 yr, 0.5 yr and 20 yr (in order of decreasing amplitudes). These are shown (in order of decreasing period) in Figure 5 (see also Table 2). The leading annual component has an amplitude (peak to trough) of 80 mm and undergoes a (longer than centennial) modulation with 10 mm amplitude. The next two components (90 and 80 yr) both have an amplitude of 40 mm, and the next three of approximately 15 mm. Taken together, they capture 95% of the total variance. The trend itself can be modeled with only three sine functions with periods 90, 60 and 20 yr.

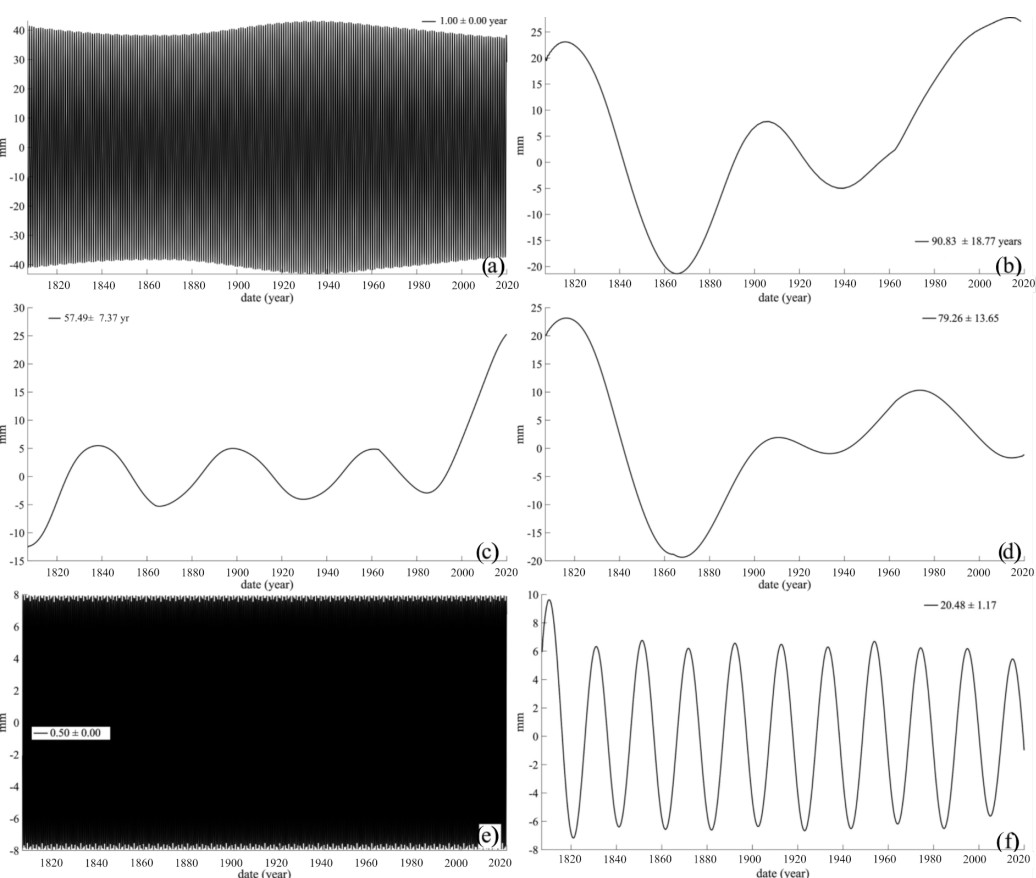

**Figure 5.** The first six most important components extracted from **GSLTG**. (**a**) First **SSA** component (1 yr) of the **GSLTG** data series of Figure 4b. (**b**) Second **SSA** component (∼90 yr) of the **GSLTG** data series. (**c**) Third SSA component (∼60 yr) of the **GSLTG** data series. (**d**) Fourth **SSA** component (∼80 yr) of the **GSLTG** data series. (**e**) Fifth **SSA** component (0.5 yr) of the **GSLTG** data series. (**f**) Sixth **SSA** component (∼20 yr) of the **GSLTG** data series.

**Table 2.** Commensurate periods of the Jovian planets [41,51]. Periods of components extracted by **SSA** from sunspot series **SSN**, **GSLl**, **GSLs** and **GSLTG** are highlighted in red.

| Planet | | Associated periods * (yr) | Courtillot et al. (2021) SSN | Church and White (2011) GSLI | Beckley et al. (2017) GSLs | This study GSLTG |
|---|---|---|---|---|---|---|
| Earth | | 0.5 <br> 1.0 | | | 0.49 ± 0.01 , 1.0 ± 0.02 | 0.5 ± 0.0 , 1.0 ± 0.0 |
| Jupiter | | 11.85 (Schwabe) <br> 5.92 | 10.56, 11.3, 13.43 <br> 5.30, 5.52 | 10.15 ± 0.58 | | |
| Saturn | | 31.44 <br> 15.72 | 35.56 <br> 15.31 | 30.96 ± 5.55 | 15.36 ± 4.70 | |
| Uranus | | 83.97 (Gleissberg) | 90.03 | | | 90.83 ± 18.77 <br> 79.26 ± 13.65 |
| Neptune | | 164.78 (Jose) <br> 83.39 | 131.02, 190.25 <br> 90.03 | | | |
| 3 { Jupiter | Saturn <br> Uranus | 9.79, 21.64 (Hale) <br> 36.06, 47.91 | 9.98, 21.42 <br> 35.56, 45.21 | 19.41 ± 1.74 | | 20.48 ± 1.17 |
| 4 { Uranus | Neptune | 40.40, 124.37 | 45.21, 131.02 | | | |
| 3 | 4 | 57.29, 67.08 | | 54.46 ± 8.68 | | 57.49 ± 7.37 |

\* from Mörth and Schlamminger (1979)

## 3. Introducing the GPS Vertical Land Motion Time Series (the [$R_{se}$(P,t) − $R_{se}$(P,$t_r$)]

As noted above, the GPS data series of VLM are all shorter than 30 years, which makes it impossible to directly combine them with the tide gauge data that are far longer on average and allow one to explore much longer periods. Unable to do better, one can at least calculate a recent (present) trend, applying a simple linear regression to the GPS

data. Furthermore, a simple linear regression of all the series shown in Figure 3 allows a comparison and combination of the gauge and VLM data: Figure 6 shows the respective histograms of the slopes of the 1548 gauges and of the GPS at the same (or close by) coastal locations (Figure 7). The median for tide gauges is +2.0 mm/yr and 92.7 % of gauges have slopes between −10 and +10 mm/yr; the median for VLM is −0.6 mm/yr and 96.0 % of the values are between −10 and +10 mm/yr. The locations and values of the tide gauges and the VLM are plotted, respectively, in Figure 7a,b. Over a large part of the coastal areas, the tide gauge signal and the VLM appear to be opposite in sign, that is, anti-symmetrical (e.g., North America, the Mediterranean region). The results of this analysis are in full agreement with the thorough study of [8], in which water rises as the land tends to subside and vice versa.

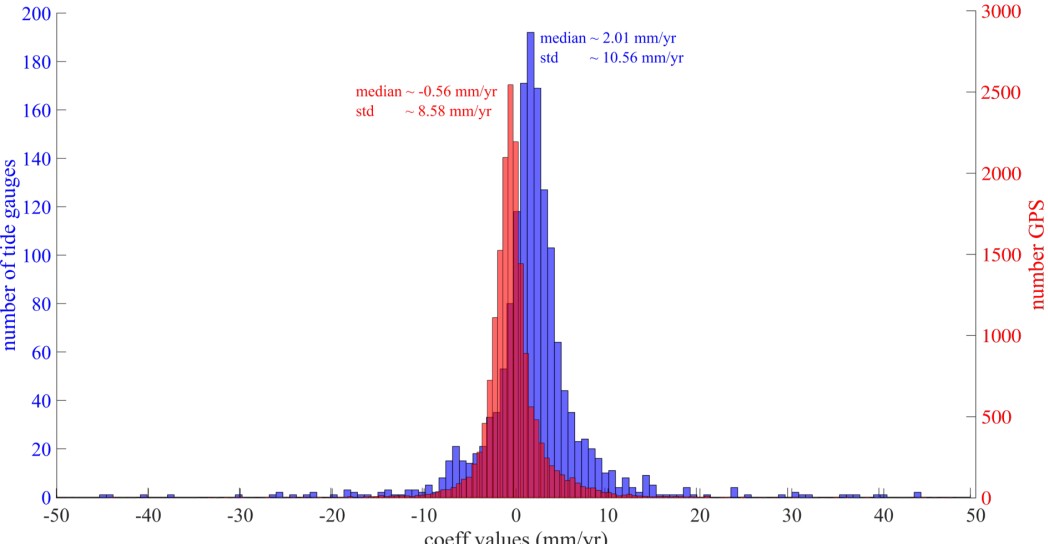

**Figure 6.** Superposition of the histogram of recent slopes of 1548 tide gauges with that of the MIDAS (GPS) data (from Hammond et al. [8]) .

If we assume that the recent (30 yr) VLM in Brest remained the same from 1807 to the present, then we can evaluate the mean sea level rise with respect to the Earth's center of mass (Equation (3)) as 1.5 mm/yr. This rough estimate is in full agreement with a series of recent studies (e.g., [8,26,57,61–64].

The MIDAS database provides the slopes of the GPS data over their duration (life time), which can range from 2 to over 20 years. The authors in [8] argue that, apart from tectonic events, these give an accurate depiction of VLM. For each tide gauge, we fit straight line segments to sea level data in the same way (though over a much longer duration, between 3 and 217 years).

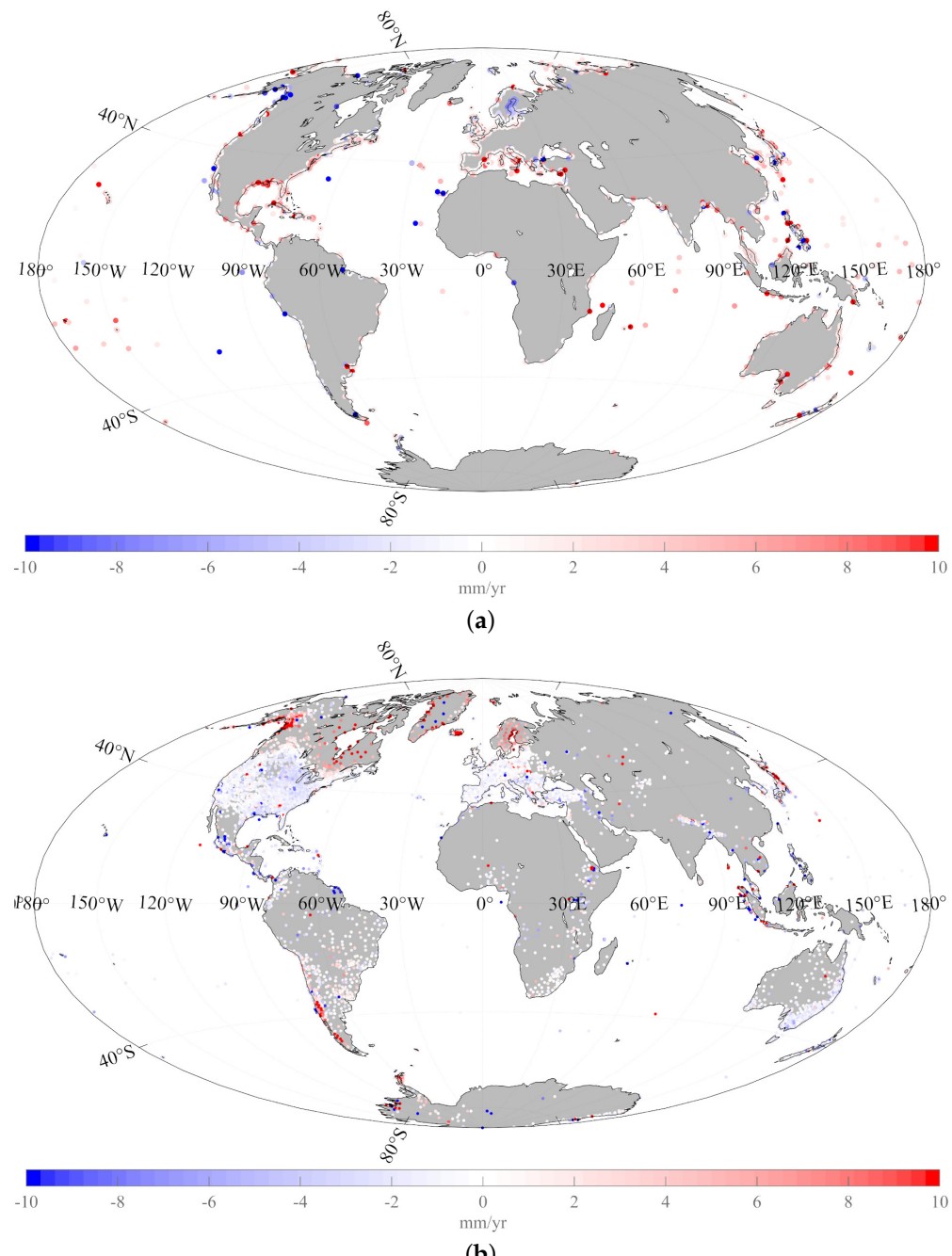

**Figure 7.** Comparison of GPS and tide gauges slopes. (**a**). The locations of tide gauges and values of the local slope of sea-level change. Color code ranges from positive (red) to negative (blue). (**b**) The locations of GPS sites and the local changes (slopes) of vertical land movement (VLM) from the MIDAS database. Color code same as in (**a**).

## 4. SSA of GSL Curves Obtained with Inclusion of Satellite Data

As far as we are interested in the long-term rise in sea level and its quasi-periodic components, but do not have VLM data over this long term, we can still use **SSA** to analyze a number of global sea-level curves available from the literature.

We select first that by [27], Figure 8. These authors estimated the rise in global average sea level from a combination of satellite altimeter data for the period 1993–2009 with coastal and is-land sea-level (tide gauge) measurements from 1880 to 2009. Variations in **GSL** as a function of time are provided by NASA (https://podaac-tools.jpl.nasa.gov/drive/files/allData/merged$_$alt/L2/TP$_$J1$_$OSTM/global$_$mean$_$sea$_$level/GMSL$_$TPJAOS$_$5.0$_$199

209$_$202008.txt accessed on 5 July 2020). The SSA of the resulting curve **GSLl** (l for long) yields a trend and components at $54.5 \pm 8.7$, $19.4 \pm 1.7$, $10.1 \pm 0.6$, $3.9 \pm 0.1$ and $31.0 \pm 5.5$ yr (Table 2 and Figure 9) in order of decreasing amplitude (or roughly, given the uncertainties trend, 60, 20, 11, 4 and 30 yr). These components are shown in Figure 09 (in order of decreasing period); their sum amounts to 89.4% of the series total variance (Figure 8). The 3.9 yr component could correspond to the second harmonic of the Schwabe cycle [38]. Note that the prominent 1 and 0.5 yr variations have been filtered out by [27].

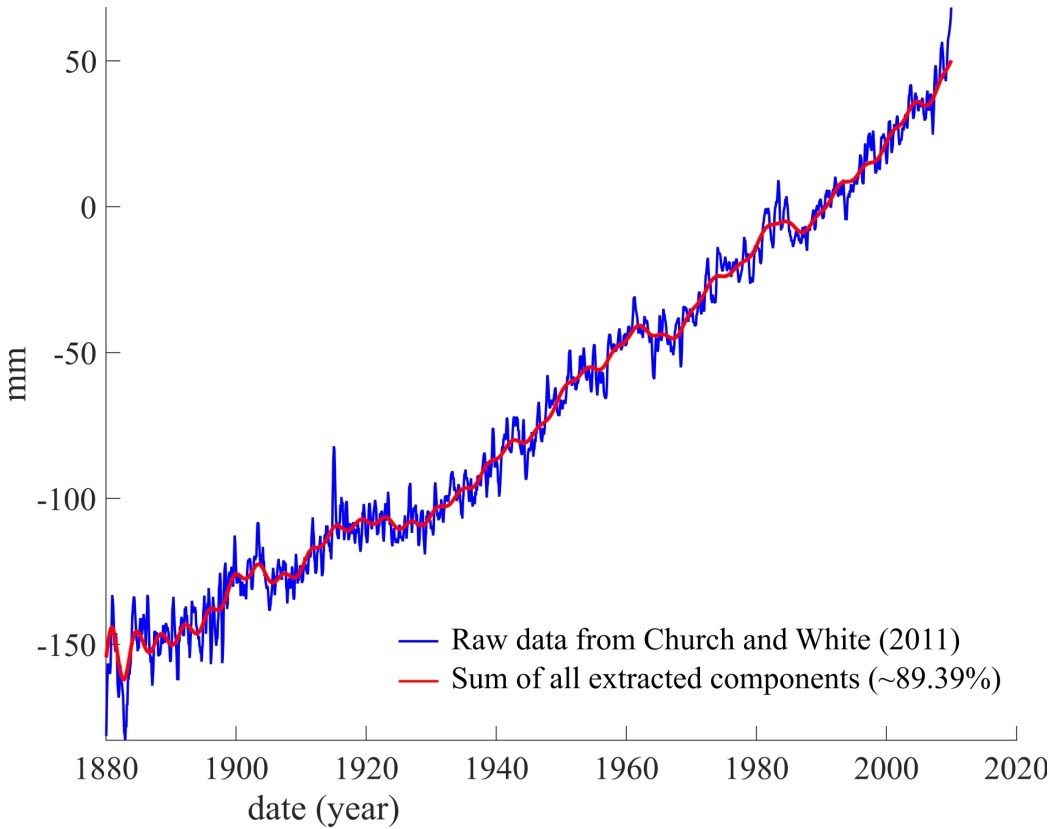

**Figure 8.** Global sea level **GSLl** curve from [27] in blue. Sum of first 6 components extracted by **SSA**, red curve.

Since the 1880–2009 **GSLl** incorporates two very different data sets (measurements from tide gauges and satellites), we have resumed the analysis for the 1993–2009 **GSLs** (s for short) based on satellite data only. For details on the structure of the data, we refer the reader to [65]. For explanations on how the data set has been created from 27 years of altimetric measurements by the successive satellites TOPEX/Poseidon (T/P), Jason-1, Jason-2 and Jason-3, we refer the reader to [66,67]. The global mean sea-level data consist in several sets of time series, some with a "Global Isostatic Adjustment applied", and some "GIA not applied". The latter (without GIA) is shown in Figure 10, from January 1993 to August 2020, with a sampling interval of 9.92 days.

The **SSA** of the resulting curve **GSLs** yields a trend and components at $1.00 \pm 0.02$, $0.49 \pm 0.01$, $3.06 \pm 0.25$, $15.4 \pm 4.7$, $1.50 \pm 0.05$ and $6.06 \pm 0.73$ yr in order of decreasing amplitude (or roughly, given the uncertainties trend, 15-20, 6, 3, 1.5, 1 and 0.5 yr). These components are shown in Figure 11 (in order of decreasing period); their sum amounts to 87.3% of the series total variance (Figure 10). The sum of the seven first components extracted by **SSA** is plotted as a red curve. The 6.1, 3.1 and 1.5 yr quasi periods could correspond to the harmonics of the Schwabe cycle (cf. [38], Table 1).

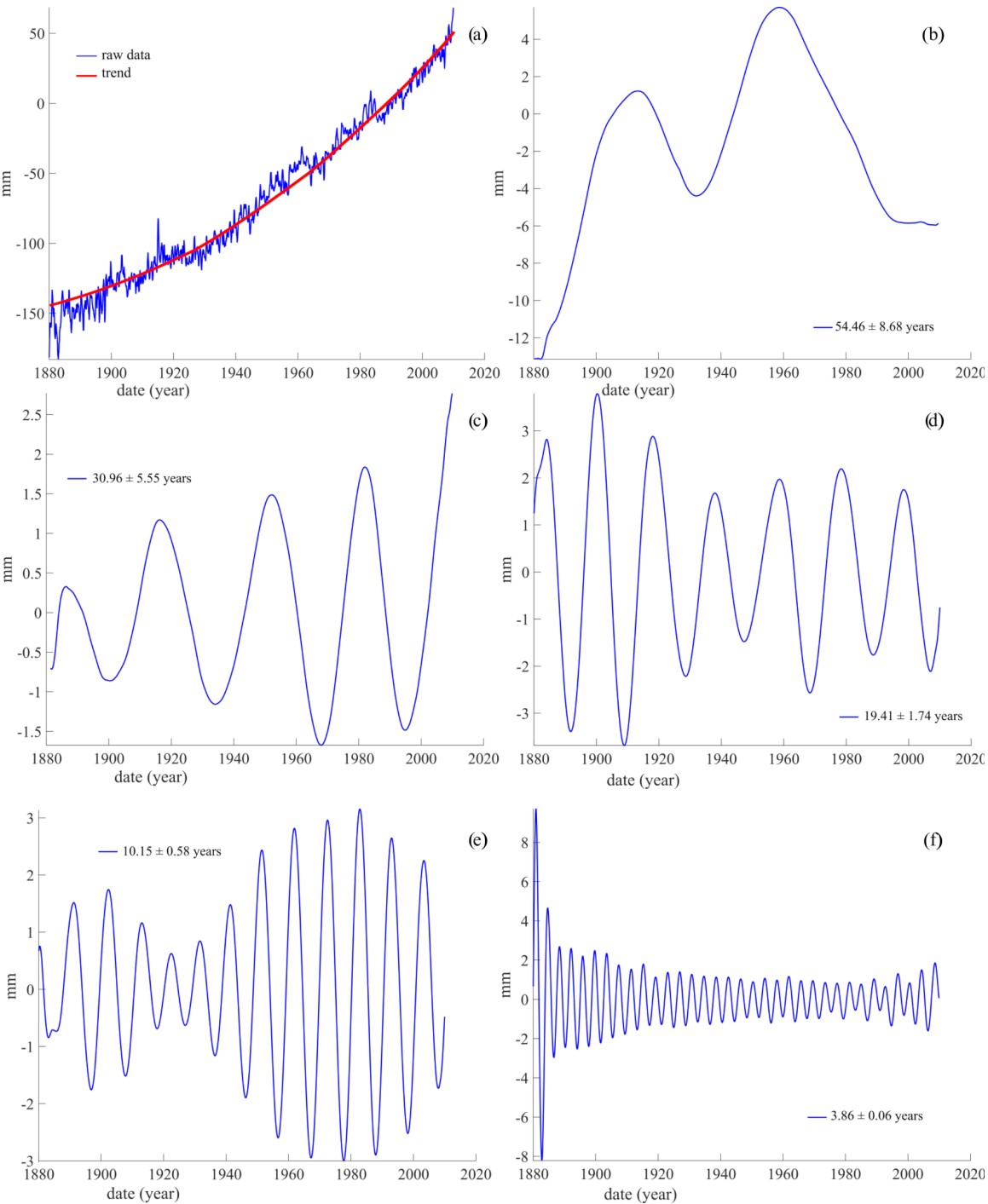

**Figure 9.** First six components extracted by **SSA** from **GSLl** data series. (**a**) The first **SSA** component (trend in red) superimposed on the **GSLl** data series. (**b**) Second **SSA** component (∼60 yr pseudo-period) of the **GSLl** data series. (**c**) Third **SSA** component (∼30 yr pseudo-period) of the **GSLl** data series. (**d**) Fourth **SSA** component (∼20 yr pseudo-period) of the **GSLl** data series. (**e**) Fifth **SSA** component (∼11 yr pseudo-period) of the **GSLl** data series. (**f**) Sixth **SSA** component (∼4 yr pseudo-period) of the **GSLl** data series.

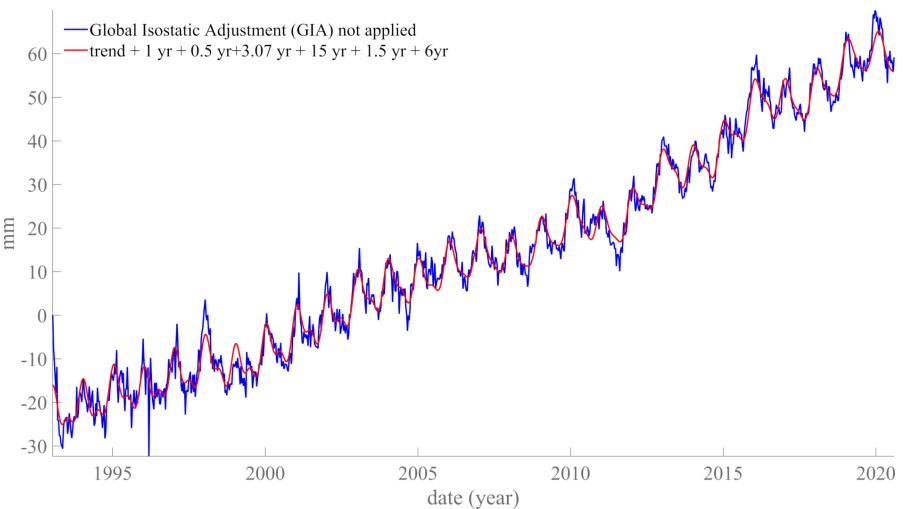

**Figure 10.** Global -sea level **(GSLs)** (blue curve). Sum of the first seven **SSA** components (red curve).

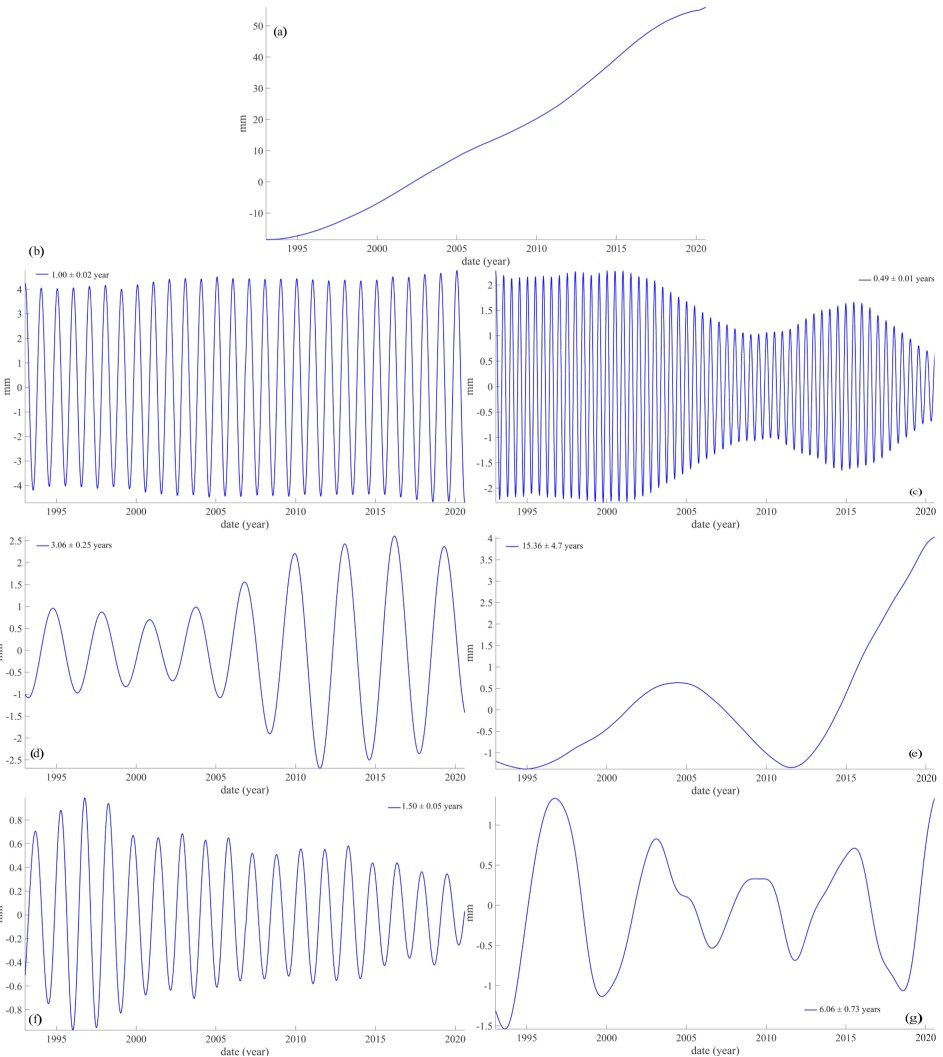

**Figure 11.** First seven components extracted by **SSA** from **GSL** data series. First **SSA** component (trend) of the **GSL** data series of Figure 10. Second **SSA** component (1 yr) of the **GSL** data series. Third **SSA** component (0.5 yr) of the **GSL** data series. Fourth **SSA** component (3 yr) of the **GSL** data series. Fifth **SSA** component (15 yr) of the **GSL** data series. Sixth **SSA** component (1.5 yr) of the **GSL** data series. Seventh **SSA** component (6 yr) of the **GSLs** data series.

## 5. Discussion

### 5.1. Components Shared by GSLl, GSLTG and SSN

Despite the fact that one is included in the other, the **GSLl** and **GSLs** series do not share many characteristics (cf. Table 2). Their trends are similar and they both have significant 1-year and 6-month components, but the two series do not share other quasi-periodic components (unless the 15-year component of **GSLs** can be considered to be the same as the 20 yr component of **GSLl**, given uncertainties). Actually, the span of the **GSLs** data is only 27 yr and should not be used to identify components with periods longer than, say, 27/2 or ∼15 yr.

In stark contrast, **GSLl** shares seven components with **GSLTG**: a trend of 60, 30, 20, 10, 1 and 0.5 yr components cf. Table 2).

Le Mouël et al. [48] and Courtillot et al. [39] have calculated the **SSA** components of the sunspot number **SSN**, a characteristic of solar activity. Four components of **SSN** are found in **GSLl** (60, 30, 20, 10 yr) and six in **GSLTG** (90, 60, 20, 30, 10, 5 yr), taking into account the uncertainties (fi. $10.15 \pm 0.58$ and $10.64 \pm 1.17$ are both considered equivalent to the **SSN** packet at 10.6 and 11.3 yr).

A puzzling observation is that the amplitude of the annual component of **GSLs** is only 8 mm peak to trough (Figure 11b), when tide gauges record a mean amplitude an order of magnitude larger (80 mm; Figure 5a). This could be due to the fact that tide gauges are located in shallow waters where wave amplitude is amplified; in any case, this is where the sea-level is relevant to human activities. The trend amplitude between 1860 and 2020 is 80 mm for **GSLTG** (Figure 4b) and more than 200 mm for **GSLl** (Figure 8). Moreover, with a **GSLTG** curve extending over two centuries, we see that the trend of the shorter series **GSLs** could actually be part of a longer cycle (in the same order as the ∼90 year Gleissberg cycle).

### 5.2. Comparison with Global Pressure GP

Thus, **SSA** reveals that **GSLl** and **GSLTG** share many characteristics that could constrain the mechanisms that control both series of sea-level change. In order to strengthen this hypothesis, one can try to find whether some other geophysical phenomenon would possess similar characteristic features with the same spectral signatures. We have searched whether the Earth's global mean pressure (**GP**) meets these requirements. A series of monthly mean atmospheric pressure (everywhere in the world) is available was of 1846 [68]. It can be accessed through the Met Office Hadley Centre (http://www.metoffice.gov.uk /hadobs/hadslp2/data/download.html accessed on 5 July 2020) website under the name HadSPL2. Following Laplace's work on the subject of the spatial and temporal stability of pressure ([10], book IV, chap 4, page 294), we built a series of monthly global pressure **GP** and submitted it to **SSA**: the first four components, i.e., the trend followed by periods of 1 yr, 6 months and ∼25 yr, account for 98% of the total variance. Thus, **GSLTG** and **GP** share three major components at ∼20/25 yr, 1 yr and 6 months. Comparisons between some features of **GSLTG** and **GP** are illustrated in Figure 12. The annual and semi-annual components match in phase and frequency and are slightly modulated with a time constant in the order of a century or more. Both series have ∼20/25 yr components that drift, one with respect to the other. Interestingly, the derivative of the trend of pressure **GP** matches the trend of **GSLTG**, suggesting a relation of the form **GSLTG** ∼ (d/dt) **GP**. Furthermore, the ratios of the amplitudes of the various **SSA** components of **GSLTG** and **GP** are approximately constant (Table 3). Sea level and pressure respond in similar ways at all time scales.

**Table 3.** Ratios of **SSA** components of **GP** to **GSLTG** with the same periods (see text).

|  | Sea Level (mm) | Pressure (hPa) | Ratio (hPa/mm) |
|---|---|---|---|
| Trend | 45 | 0.8 | 0.019 |
| ∼20–30 yr | 18 | 0.45 | 0.025 |
| 1 yr | 80 | 1.6 | 0.020 |
| 0.5 yr | 16 | 0.3 | 0.019 |

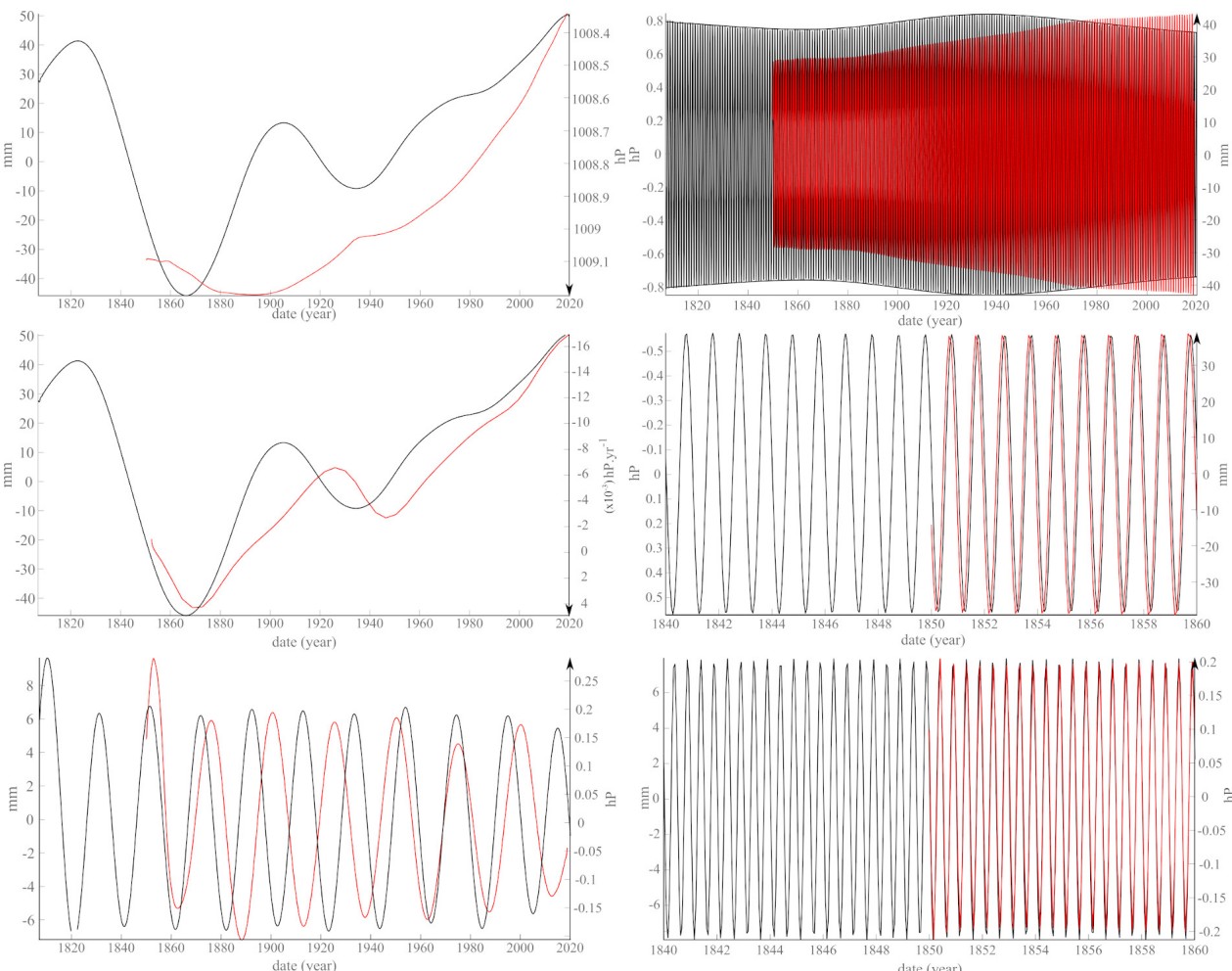

**Figure 12. Top left**, the superposition of the trends of **GSLTG** (in black) and **GP** (in red). **Middle left**, the superposition of the trend of **GSLTG** (in black) and the first derivative of the trend of **GP** (in red). **Bottom left**, the ∼20 yr components of GP (red) and GSLTG (black). **Top right** and **middle right** (a zoom), the annual oscillations of **GSLTG** (black) and **GP** (red). **Bottom right**, the semi-annual cycles of **GP** (red) and **GSLTG** (black).

*5.3. Comparison with the Mean Motion of the Rotation Pole (RP)*

It is generally accepted (e.g., [17]) that the mean motion of the rotation pole (**RP**) and the mean sea-level (be it **GSLTG** or **GSL**) belong to the same family because of the reorganization of surface masses due to pole motion. Because the Liouville–Euler equations are a linear differential system of second order, one understands the resemblance between **RP** and both **GSLTG** or **GSL**, at least during longer periods. Can the analysis of the **RP** series bring additional light to the question of the forcing of sea level (we refer the reader to earlier work on polar motion by [31,32,51])?

The series of data describing the rotation pole coordinates (**RP**) is maintained by the International Earth Rotation and Reference Systems Service (**IERS**, https://www.iers.org

/IERS/EN/DataProducts/EarthOrientationData/eop.html accessed on 5 July 2020). Two series of measurements of pole coordinates $m_1$ and $m_2$ are provided by **IERS** under the codes EOP-C01-IAU1980 and EOP-14-C04. The first one runs from 1846 to 1 July 2020 with a sampling rate of 18.26 days, and the second runs from 1962 to 1 July 2020 with daily sampling. Figure 13 displays the trends (top) and derivatives (bottom) of the rotation pole (**RP**) coordinates $m_1$ and $m_2$ together with those of **GSLTG** (1807–2020). There is a suggestive anti-correlation between the trend of **GSLTG** and the derivative of the trend of $m_2$ (**GSLTG** $\sim$(d/dt) **RP**), with the former leading the latter by several decades (Figure 13, lower left). This behavior was noted for the Brest tide gauge by [32]. This is yet another line of observational evidence that the sea-level curve based on tide gauges has an underlying physical mechanism.

The three main components of pole motion **RP**, i.e., the Markovitch drift, the Chandler free oscillation (that has never yet been detected in sea level) and the annual forced oscillation, carry more than 75% of the signal's total energy (variance) [31,51,53]. Ref. [31] also showed the presence of other components at 22 yr (Hale), 11 yr (Schwabe) and a 5.5 yr harmonic. The order of magnitude of these solar components is $10^{-12}$ to $10^{-14}$ rad.sec$^{-1}$, that is 1–4 orders of magnitude smaller than the main components ($10^{-11}$ to $10^{-10}$ (rad· sec)$^{-1}$). In [26,39,51], it has been shown that the derivative of the Markovitch component includes the 90 yr (Gleissberg) cycle as its trend. Ref. [51] identified some other components that also appear in the **SSN** sunspot series [39]. We limit ourselves to the shared (quasi-) periods of 90, 22, 11, 5.5, 1.4 and 1 year.

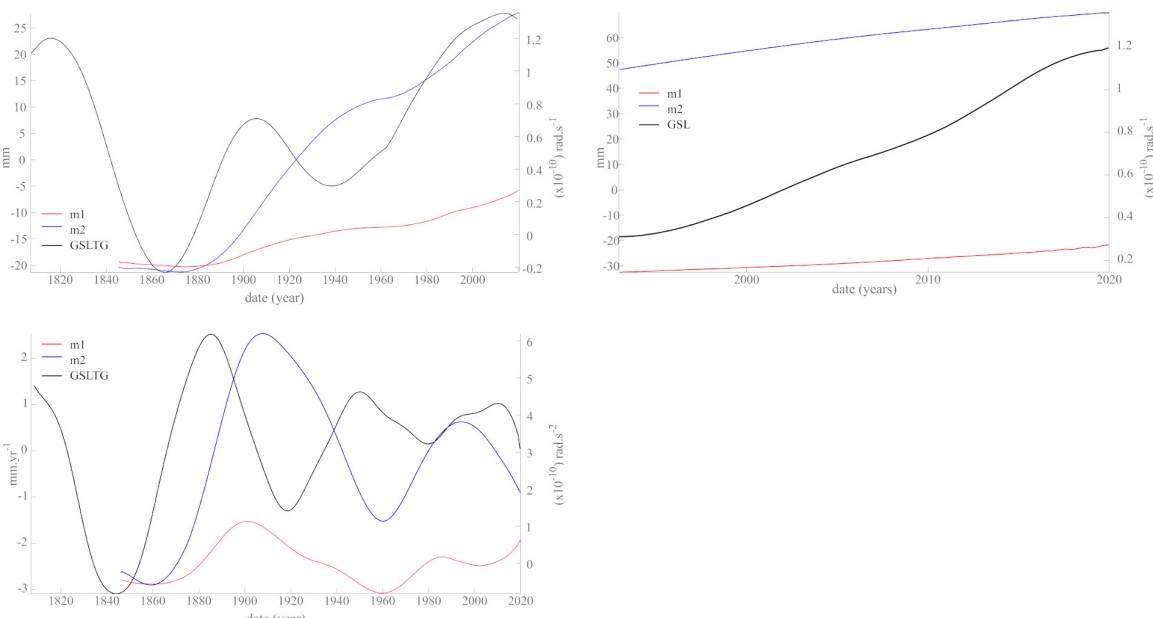

**Figure 13.** Trends (**top**) and their derivatives (**bottom**) of the rotation pole (**RP**) coordinates $m_1$ (red) and $m_2$ (blue) compared with those for **GSLTG** (**left**; 1807–2020).

*5.4. The Liouville–Euler System and Solar Components in Sea Level*

The Liouville–Euler system of equations can be written (*cf.* [9]; chapters 3 and 4):

$$(1/\sigma_r)\frac{d\mathbf{m}}{dt} + \mathbf{m} = \mathbf{f}$$
$$\frac{dm_3}{dt} = f_3 \tag{4}$$

$\frac{d\mathbf{m}}{dt} = m_1 + i*m_2$, where $(m_1,m_2)$ are the coordinates of motion of the Earth's rotation pole (**RP**) and $\mathbf{f} = f_1 + i*f_2$ and $f_3$ are the excitation functions. These are explicitly written in [9], chapter 4, system 4.1.1, page 47. These functions belong to three distinct families: masses,

motions (of masses), and imposed forces (torques). Parameters ($m_1$,$m_2$,$m_3$) provide a global scalar for pole motion, and the excitation functions are also global. Because coupled system (4) is linear, any periodic component found in ($m_1$,$m_2$,$m_3$) data/observations must also be present in ($f_1$,$f_2$,$f_3$).

If system (4) does apply, since variations in sea-level are the motion part of the excitation functions, and since the Gleissberg cycle (90 yr) and annual components have been found, one should also find 1.4 yr component. In the same way, Moreira and Cazenave [69] identified an 11 yr oscillation, Jevrejeva et al. [54] a 22 yr oscillation and Chambers [55] another one at 60 yr (solar, *cf.* [39]) with weak amplitude in polar motion (*cf.* [51,53]). Thus, cycles that are commonly associated to the Sun are also detected in sea level, though they are a very weak component of polar motion.

*5.5. Planetary Forcings*

Laplace predicted that all terrestrial masses should be under the influence of other celestial bodies, at much longer periods (though he did not have the data to demonstrate it experimentally). For these longer durations, planets were the potential culprits. If one studies durations that are short compared to a decade, the 11 yr component will be included in the trend. In the same way, if one looks at the variations shorter than 5 days, the 28 day lunar cycle will be included in the trend. This is particularly true for fluid envelopes. In [39], we showed that sunspots, which move on the external fluid envelope of the Sun, include components with periods that can be linked to planetary ephemerids (commensurable periods of the Jovian planets), as proposed by [41], namely ∼ 165 yr (José cycle, associated with Neptune), ∼90 yr (Gleissberg cycle, associated with Uranus), ∼30 yr (Saturn), ∼ 22 yr (Hale cycle, a Jupiter/Saturn commensurability), ∼ 11 yr (Schwabe cycle, associated with Jupiter), etc. See Lopes et al. ([70]) for more details.

The solid Earth acts as an integrator (e.g., [9], Chapter 3) at the longer periods (e.g., [51]), but reacts almost instantaneously to the shorter periods (e.g., [71]). The effects are large in the former case, but less so in the latter. As pointed out by [10], Book IV, Chapters 4 and 5, because the atmosphere and oceans are fluids with low viscosity, they actually behave as solid envelopes at the longer periods. This is for instance the reason why the annual oscillations observed on tide gauges (Figure 11) have similar amplitudes. To this response is added the response described by the Liouville–Euler equations. It is used for instance by [17], who considered that there is a reorganization of surface masses when the inclination of the Earth's rotation axis evolves in space. External forces influence polar motion and this should also be seen in sea-level and pressure. We do find the signatures of these influences in the trends (directly or through their derivatives) the most energetic components. The apparent rise of the sea-level is similar to the pattern of polar motion: this is why, like others, we find this funnel pattern of tide gauges. The mass of the system is constant; this agrees with the fact that the pressure variation remains constant and linked with the sea-level. Under the hypothesis of common external forcing (common to the solid Earth and its fluid envelopes) this would explain why the derivative of the trend of **GP** (with the sign reversed) and the trend of **GSLTG** match quite well (Figure 12, middle left).

## 6. Summary and Concluding Remarks

In the present paper, we have first analyzed variations in the geoid, using tide gauge data for the time span 1807–2020 (**GSLTG**) and variations in the global sea level combining tide gauges and satellite data (**GSLl** [27]); then, with much better spatial coverage but a much more restricted time span (1993-2020), satellite-only data (**GSLs**, [65]). Our main goal has been to determine the trends and successive periodic or quasi-periodic components of these time series, using the **SSA** method (**SSA**) that we have used, with interesting results, in a series of previous papers [26,31,39,47,48,51].

Since satellite data are first corrected using tide gauges, we analyzed the tide gauge data at 71 gauge stations separately (series **GSLTG**). We see all kinds of behavior, with increasing, stable or decreasing mean values (trend), giving the complete data set a funnel

shape. The resulting **GLSTG** mean oscillates around an almost constant value. From 1860 to 2020, the trend increases by 90 mm, or a contribution to the mean rise rate of 0.56 mm/yr.

Our observations are compatible with previous results also based on tide gauge data but using different methods: [19] Figure 02, [72] Figures 9.1 and 9.2 and [24] Figure 03 show the funnel-shaped distribution of the data with values in agreement with **GSLTG**.

**SSA** analysis of the **GSLTG** series identifies components with celestial periods already found when analyzing the Brest tide gauge data [26]. We have shown (Table 3) that the ratios of amplitudes of **SSA** components (trend, 20–30 yr, 1 yr, 0.5 yr) of **GSLTG** vs. global pressure **GP** are almost constant (at 0.02 hPa/mm). This observation supports the idea that tide gauge data contain significant information on the physical links between mean pressure and (geoid) sea-level.

The longer (annual to multi-decadal) periods we encountered in the **GSLTG** and **GSLl** sea-level curves in the present study have already been encountered in a number of geophysical and heliophysical time series. These periods (or quasi-periods) are ∼80–90, 60, 30, 20, 10–11, and 4–5 years. They can be compared to the commensurable periods of the Jovian planets acting on the Earth and Sun as proposed by [41]. The combination of the revolution periods of Neptune (165 yr), Uranus (84 yr), Saturn (29 yr) and Jupiter (12 yr) and several commensurable periods makes additional pseudo-cycles of 60 and 20 yr appear in sunspots [32,39] as well as in a number of terrestrial phenomena [31,32,41,45,46,49,51,58], particularly in sea level [55,73–76]. It is therefore not a surprise to find Jovian periods in **GSLTG**. Even trends could be part of commensurate cycles longer than the time span over which the data are available.

The (often joint) periods encountered in the sea-level changes, the motion of the rotation pole **RP** and the evolution of global pressure **GP** and the relations between some of their trends (as determined using **SSA**) suggest that there may be physical links and causal relationships between these geophysical phenomena and the time series of observations available. The ubiquitous presence of many common components in the variations of many natural phenomena (that a priori might seem largely unrelated) has led us to return to the general forcing envisioned by [10] and to the full theory he developed. In their 1799 Treatise of Celestial Mechanics, Laplace derived the Liouville–Euler partial differential equations that describe the rotation and translation of the rotation axis of any celestial body, and showed that the only thing that influences the rotation of celestial bodies is the action of other celestial bodies. Laplace emphasized that one must consider the orbital kinetic moments of all planets in addition to gravitational attractions and concluded that the Earth's rotation axis should undergo motions with components that carry the periods or combinations of periods of the Sun, Moon and planets (particularly Jovian planets).

In [51,53], we tested Laplace's theoretical results using observations that have accumulated since their time. In [39], we gave evidence of the driving influence of the planets (mainly the Jovian planets) on the Earth's rotation axis [51], and on solar activity, through exchanges of angular momentum. In the present paper, we have extended the analysis to changes in global sea-level, as measured by tide gauges (**GSLTG**), and we have also shown that there were several shared periodic components with global pressure **GP** and the Earth's rotation axis **RP**.

Laplace's treatment shows that the kinetic moments of planets act both directly on the Earth and on the Sun's fluid external layers, and perturb its rotation; hence, its revolution and eventually the Earth's axis of rotation **RP**. The **SSA** analysis of the envelopes of the derivatives of the three first polar motion components yields a number of additional periods that belong to the series of commensurable periods, among which 70 yr, 60 yr (Saturn; also found in global temperatures and oceanic oscillations), 40 yr (a commensurable revolution period of the four Jovian planets), 30 yr (Saturn) and 22 yr (Jupiter and Solar). The same is true for the first three **SSA** components of sunspot series **SSN** (trend or Jose 175 year cycle, linked to Neptune; 11 yr Schwabe cycle, linked to Jupiter; and 90 yr Gleissberg cycle, linked to Uranus). Almost all the (quasi-) periods found in the **SSA** components of sea-level (**GSLl** and **GSLTG**), global pressure (**GP**) and polar motion (**RP**) of their modulations and

of their derivatives can be associated with Jovian planets (Table 3). These complement the list of **SSA** quasi-periodic and periodic components that are found in **GSLl**, **GSLTG**, **GP**, **RP** and **SSN** (90, 60, 30, 20, 10, 5, 1 and 0.5 years) and can all be associated with planetary forcings, as envisioned by Laplace. In particular, planetary forcing factors are likely causally responsible for many of the components of sea-level variations as measured by tide gauges. It is of particular interest to search for high-quality data series on longer time intervals, that can allow one to test whether the trends themselves could be segments of components of even longer periodicities (e.g., 175 yr Jose cycle). In any case, the first **SSA** components of the series analyzed in this paper comprise a large fraction of the signal variance: 95% for the first six components of **GSLTG**, 89% for the first six components of **GSLl**, 87% for the first six components of **GSLs**, 98% for the first four components of **GP** and 75% for the first three components of **RP**. It is clear that one should attempt to physically model these series with this set of periods (**SSA** components) before trying to invoke alternate sources (forcing functions).

**Author Contributions:** V.C., J.-L.L.M., F.L. and D.G. contributed to conceptualization, formal analysis, interpretation and writing. All authors have read and agreed to the published version of the manuscript.

**Funding:** This research was supported by the Université de Paris, IPGP and the LGL-TPE de Lyon.

**Data Availability Statement:** The used data are freely available at the following address: Permanent Service of Mean Sea Level: https://www.psmsl.org/data/obtaining/complete.php accessed on 5 July 2020; Median Interannual Difference Adjusted for Skewness: http://geodesy.unr.edu/velocities/ accessed on 5 July 2020; NASA: https://podaac-tools.jpl.nasa.gov/drive/files/allData/merged$_$alt/L2/TP$_$J1$_$OSTM/global$_$mean$_$sea$_$level/GMSL$_$TPJAOS$_5.0$_$199209$_$202008.txt accessed on 5 July 2020; Met Office Hadley Centre: http://www.metoffice.gov.uk/hadobs/hadslp2/data/download.html accessed on 5 July 2020; International Earth Rotation and Reference Systems Service: https://www.iers.org/IERS/EN/DataProducts/EarthOrientationData/eop.html accessed on 5 July 2020.

**Acknowledgments:** We thank the anonymous reviewers for their helpful comments on the manuscript.

**Conflicts of Interest:** The authors declare that they have no known competing financial interest or personal relationships that could have appeared to influence the work reported in this paper.

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
