# Peer review of "On Sea-Level Change in Coastal Areas"

_jmse, doi:10.3390/jmse10121871_

Round 1

Reviewer 1 Report

jmse-2041012-peer-review-v1

This is a sound manuscript. It is well-researched and nicely written.

I only have minor suggestions as follows:

LN 16. “data series with longer spans.” Is it a feasible suggestion? Because this type of data is complicated to acquire. However, the text could be complemented with an example like this: “data series with longer time spans such as…”

LN 85. The acronym SSA has been described before. Please check that all acronyms are not repeated throughout the manuscript.

Figures 4, 7, and 15 should be tables, not figures. And is there a particular reason regarding the red coloration within the text?

Figures 10 & 11 present no data from 1820 to 1880. Is there a particular reason for such a blank region? Could it be cut and started in 1880?

Subfigures within figures 6, 11, and 13 are pretty big. I prefer the format of figure 14. Could it be possible to stack figures 6, 11, and 13 just like figure 14?

There are some inconsistencies in using the serial comma throughout the manuscript. e.g., LN 351 vs. 353.

Missing acknowledgments.

Author Response

Please find our comments and responses in the uploaded pdf file

Reviewer 2 Report

In general, this manuscript is very interesting for the variations in sea-level, based on tide gauge data (GSLTG) and on combining tide gauges and satellite data (GSLl) are subjected to singular spectrum analysis (SSA), to determine their trends and periodic or quasi-periodic components. The research would be of interest to search for data series with longer time spans, which could allow one to test whether the trends themselves could be segments of components with still longer periodicities.

 Specifically, there are somewhere to be corrected or improved as below:

 1)     Line2, please check all the text about “GSLI” or “GSLl”? Is there any confusion between upper case letter “I” and lower case letter “l”?

 2)     Line14-18, “Almost all the periods found in the SSA components of sea-level (GSLl and GSLTG), global pressure (GP) and polar motion (RP), of their modulations and their derivatives can be associated with the Jovian planets.”

It can be understood that the sea level periods change is related to the planetary revolution periods. In a relatively short time, will the sea level change caused by global warming affect these periods? In addition, does the correlation between the periodic change of sea level and the planetary revolution periods have more practical significance for future sea level research?

3)     Line 69-70, The authors demonstrated the raw data of all the 1548 sites. But in fact, it only used the data from 31 sites. My suggestion is to delete the content that doesn't seem very meaningful.

4)     Figure 3, The name of Figure 3 is “Superposition of data series from the 31 tide gauges listed in Table fig. 4”. Should exchange the position of Figure 3 and Figure 4 in order to make reading more smoothly?

5)     Figure 4, In view of the different establishment time of different tide gauges, and the lack of measurement period, please explain the time resolution and the time scale of the data.

6)     Line 125-126, “…and for reasons that will appear in the discussion.” Please briefly summarize the reason here first.

7)     Line 141-142, “Taken together, they capture 95% of the total variance.” Are all these components significant?

8)     Line 142-143, “The trend itself can be modeled with only 3 sine functions with periods 142

160, 90 and 30 yr.” I can’t understand why the trend itself can be modeled with only periods 160, 90, 30yr? Please explain it.

9)     Figure 6 (c), Please correct the mathematical symbols in the figure.

10)   Figure 6 (e), Please make the picture more enjoyable, such as adjusting the coordinate range.

11)   Figure 6, “The first sixt” whether it means "the first six"?

12)   Figure 7, Have the SSN, GSLI, GSLS data been calculated in the author's previous research? Please indicate the source

13)   Figure 8, Please add the legends to explain what the red and blue means and check “(GPS) data (from [8]”.

14)   Line 178, Please correct “(Table fig. 7 and Figure??)”.

15)   Line 180, Which picture does “Figure 09” represent?

16)   Figure 10&11, The abscissa should start from 1880?

17)   Figure 14, Please mark the serial number in the figure.

18)   Figure 12, Please mark the serial number in the figure.

19)   Line 295, “the 28 days”.

Author Response

(The authors gave the same response as above.)
